# Targeting OLFML3 in Colorectal Cancer Suppresses Tumor Growth and Angiogenesis, and Increases the Efficacy of Anti-PD1 Based Immunotherapy

**DOI:** 10.3390/cancers13184625

**Published:** 2021-09-15

**Authors:** Jimmy Stalin, Beat A. Imhof, Oriana Coquoz, Rachel Jeitziner, Philippe Hammel, Thomas A. McKee, Stephane Jemelin, Marine Poittevin, Marc Pocard, Thomas Matthes, Rachid Kaci, Mauro Delorenzi, Curzio Rüegg, Marijana Miljkovic-Licina

**Affiliations:** 1Department of Pathology and Immunology, University of Geneva Medical School, Rue Michel Servet 1, CH-1211 Geneva, Switzerland; beat.imhof@unige.ch (B.A.I.); philippe.hammel@unige.ch (P.H.); stephane.jemelin@unige.ch (S.J.); marine.poittevin@hotmail.fr (M.P.); Marijana.Licina@unige.ch (M.M.-L.); 2Department of Oncology, Microbiology and Immunology, Faculty of Science and Medicine, University of Fribourg, Chemin du Musée 18, PER17, CH-1700 Fribourg, Switzerland; oriana.coquoz@unifr.ch (O.C.); curzio.ruegg@unifr.ch (C.R.); 3Medicity Research Laboratory, University of Turku, Tykistökatu 6A, 20520 Turku, Finland; 4Bioinformatics Core Facility, SIB Swiss Institute of Bioinformatics, CH-1015 Lausanne, Switzerland; Rachel.Jeitziner@sib.swiss (R.J.); Mauro.delorenzi@unil.ch (M.D.); 5Division of Clinical Pathology, Geneva University Hospital, Rue Michel Servet 1, CH-1211 Geneva, Switzerland; thomas.a.mckee@hcuge.ch; 6CAP Paris-Tech, Université de Paris Diderot, INSERM U1275, 49 Boulevard de la Chapelle, CEDEX 10, F-75475 Paris, France; marc.pocard@gmail.com (M.P.); rachid.kaci@aphp.fr (R.K.); 7Department of Oncologic and Digestive Surgery, AP-HP, Hôpital Lariboisière, 2 Rue Ambroise Paré, CEDEX 10, F-75475 Paris, France; 8Department of Oncology, Hematology Service, Geneva University Hospital, Rue Michel Servet 1, CH-1211 Geneva, Switzerland; thomas.matthes@unige.ch; 9Department of Diagnostics, Clinical Pathology Service, Geneva University Hospital, Rue Michel Servet 1, CH-1211 Geneva, Switzerland; 10Translational Research Centre in Oncohaematology, University of Geneva Medical School, Rue Michel Servet 1, CH-1211 Geneva, Switzerland; 11Department of Anatomopathology, AP-HP, Hôpital Lariboisière, 2 Rue Ambroise Paré, CEDEX 10, F-75475 Paris, France; 12Department of Oncology, University Lausanne, CH-1011 Lausanne, Switzerland

**Keywords:** colorectal cancer, tumor development, tumor angiogenesis, blood vessel pericyte coverage, checkpoint inhibitors

## Abstract

**Simple Summary:**

Tumor vascularization promotes tumor growth and is intimately connected to immune system function. Despite efforts to use antiangiogenic therapies, their clinical effects are less pronounced than expected. Compensatory angiogenic responses and resistance to treatment are responsible for these limited therapeutic effects. Inhibition of OLFML3 suppresses the growth of colorectal cancer in preclinical models. Our study identified OLFML3 as a key regulator of angiogenesis, lymphangiogenesis, pericyte coverage, tumor-associated macrophage recruitment, and enhanced NKT lymphocyte recruitment associated with its antitumor effects. We also highlight that OLFML3 antibodies increase the efficacy of anti-PD-1-based therapies. These results are in agreement with the high expression of OLFML3 observed in colorectal carcinoma patients associated with shorter relapse-free survival, higher grade, and angiogenic microsatellite stable (CMS4) subtype. Clinically, high OLFML3 expression correlates with reduced disease-free survival in human colorectal cancer patients, suggesting a potential role as a therapeutic target.

**Abstract:**

The role of the proangiogenic factor olfactomedin-like 3 (OLFML3) in cancer is unclear. To characterize OLFML3 expression in human cancer and its role during tumor development, we undertook tissue expression studies, gene expression analyses of patient tumor samples, in vivo studies in mouse cancer models, and in vitro coculture experiments. OLFML3 was expressed at high levels, mainly in blood vessels, in multiple human cancers. We focused on colorectal cancer (CRC), as elevated expression of OLFML3 mRNA correlated with shorter relapse-free survival, higher tumor grade, and angiogenic microsatellite stable consensus molecular subtype 4 (CMS4). Treatment of multiple in vivo tumor models with OLFML3-blocking antibodies and deletion of the Olfml3 gene from mice decreased lymphangiogenesis, pericyte coverage, and tumor growth. Antibody-mediated blockade of OLFML3 and deletion of host Olfml3 decreased the recruitment of tumor-promoting tumor-associated macrophages and increased infiltration of the tumor microenvironment by NKT cells. Importantly, targeting OLFML3 increased the antitumor efficacy of anti-PD-1 checkpoint inhibitor therapy. Taken together, the results demonstrate that OLFML3 is a promising candidate therapeutic target for CRC.

## 1. Introduction

Tumor angiogenesis is the process of new blood vessel formation, which drives cancer growth and progression [1]. Angiogenesis is induced and regulated by numerous factors, including vascular endothelial growth factor (VEGF), placental growth factor (PLGF), fibroblast growth factor (FGF), and bone morphogenetic protein 4 (BMP-4) [2,3,4,5,6]. Bevacizumab, a monoclonal antibody targeting VEGF-A, was the first clinical antiangiogenic agent approved in colorectal cancer therapy [7,8,9]. However, bevacizumab only provides transient survival benefits, and in some instances, it shows low additional therapeutic effects relative to chemotherapy alone [10]. The transient nature of the antitumor effects of bevacizumab suggests the existence of alternative mechanisms of angiogenesis or events that compensate for VEGF-A inhibition. These include expression of other angiogenic factors produced constitutively (intrinsic resistance) or during anti-VEGF-A treatment (acquired resistance) and/or development of adaptive mechanisms by the tumor cells themselves, including resistance to hypoxia, increased cell motility and survival, and escape from host immune cells [11]. Therefore combined therapies have been suggested or are in use [12].

Pericytes are thought to confer resistance to anti-VEGF treatment; as such, they are increasingly considered additional targets for anticancer therapies. Pericytes, together with smooth muscle cells, surround the endothelium and contribute to the structure of the vascular wall. They are involved in both physiological (e.g., vessel stability) and pathological (e.g., increased permeability) vascular processes [13,14]. Pericytes express multiple markers, including neuron glial-2 (NG2), alpha-smooth muscle actin (α-SMA), and platelet-derived growth factor (PDGF) receptor β (PDGFRβ) [15]. The PDGF-B/PDGFRβ signaling pathway promotes pericyte proliferation, migration, survival, and attachment to endothelial cells during angiogenesis [16]. Impaired PDGF-B/PDGFRβ signaling results in failure of pericyte recruitment and reduced pericyte coverage, which in turn leads to endothelial hyperplasia, abnormal vascular morphogenesis, and formation of microaneurysms [17,18].

Human OLFML3 is a matricellular protein with proangiogenic properties belonging to the family of olfactomedin-domain-containing proteins [19]. Mouse Olfml3 is expressed in the presumptive vasculogenic regions of embryos during development [20], whereas expression in adult animals is largely limited to tissues undergoing remodeling [21]. Moreover, as we reported previously [22], Olfml3 is expressed by endothelial cells and pericytes within the tumor stroma, where it promotes vascularization and growth of mouse Lewis lung carcinoma (LLC1). Olfml3 serves as a scaffold protein that recruits bone morphogenetic protein 1 (BMP1) to its substrate chordin, a BMP antagonist [23]. Both mouse Olfml3 and human OLFML3 interact with BMP4 [22], a proangiogenic factor involved in tumor cell migration and invasion [24]. Interestingly, targeting Olfml3 by polyclonal antibodies inhibits tumor growth in a mouse model of lung carcinoma [22]. Recent evidence suggests that deletion of the Olfml3 gene from mice alters developmental angiogenesis by modulating the PDGF-dependent recruitment of pericytes, leading to in utero hemorrhage and partial lethality [25]. However, even though OLFML3 is a candidate target for tumor targeting [26], little is known about its relevance to human cancer or its therapeutic potential.

Here, we examined the expression of OLFML3 mRNA and protein in human tumors, including colorectal cancer (CRC) and lung carcinoma. We used recombinant monoclonal antibodies specific for OLFML3 to block tumor angiogenesis, lymphangiogenesis, and pericyte coverage in mouse models and monitored the effects on tumor growth. We also tested whether compensatory proangiogenic responses occur. Next, we investigated whether blocking the expression of OLFML3 correlates with the recruitment of tumor-associated macrophages (TAMs) and NKT cells to human CRC. Finally, we asked whether the antitumor effect of combined treatment with anti-OLFML3 and anti-PD-1 antibodies is more efficient than monotherapy. In parallel, we investigated whether the antitumor effects of anti-OLFML3 antibodies were reproduced by the deletion of Olfml3 from mice.

## 2. Materials and Methods

### 2.1. Human Tumor Samples

Samples were obtained from the human Tissue Bank Bern (TBB) in accordance with the Human Research Act (Humanforschungsgesetz, HFG) and international guidelines (ethics approval number KEK 200/2014). Tumor samples of reviewed cases were analyzed by next-generation tissue microarrays (ngTMAs) at the Institute of Pathology, University of Bern [27]. Regions of interest were annotated on H&E-stained and digitally scanned slides and subsequently punched (0.6 mm diameter) with a TMA tool on an automated tissue microarrayer. Adjacent sections were stained with a peptide A+B-purified polyclonal rabbit anti-OLFML3A+B antibody [22]. Immunohistochemically (IHC) stained sections were scanned, and OLFML3 expression analyzed.

For the PDX experiments, samples were from a PDX tumor collection of primary human colorectal cancers obtained with patients’ informed consent [28]. Briefly, primary tumors were engrafted subcutaneously into CB17-SCID mice (Charles River Laboratories, Wilmington, MA, USA). Xenografts were collected when they reached a volume of 0.15–0.3 cm^3^ and frozen at −80 °C until further use. Total RNA and proteins were extracted from a total of 27 different xenograft samples using the NucleoSpin RNA/Protein extraction kit (Macherey-Nagel, Düren, Germany) for analysis.

To analyze bevacizumab-treated tumor samples, tumor tissues obtained from patients with stage 4 naïve tumors, treated or not (controls) with bevacizumab undergoing surgery, were employed. The protocol was approved by the institutional ethical committee of Lariboisiere Hospital (Paris, France), and all patients provided informed consent. Formalin-fixed paraffin-embedded (FFPE) tissues were sectioned and stained with H&E to confirm the presence of viable tumor cells, and total RNA and protein were extracted from sections using the ExpressArt FFPE Clear RNAready kit (Amsbio, Abingdon-on-Thames, UK).

### 2.2. Bioinformatics Analysis of Colorectal Tumor Datasets

Relapse-free and overall survival curves were generated using the Kaplan–Meier method in the R library ‘survminer’ (Drawing Survival Curves software, ‘ggplot2’ R package, version 0.4.4.); patients were stratified according to the expression of OLFML3 relative to the median value. Expression and clinical data were taken from the Cancer Genome Atlas Research Network (https://www.cancer.gov/tcga, accessed on 1 April 2019 and the GEO repository entry GSE39582. Expression of OLFML3 in different subgroups of patients was compared using *t*-tests with equal variance.

Pearson’s correlation and Spearman’s rank correlation coefficients were calculated to analyze the relationships between OLFML3 expression and that of genes of interest. Tested gene sets included angiogenesis genes (gene ontology, GO:0001525), genes known to be positively (GO:0045766) or negatively (GO:0016525) associated with angiogenesis, genes associated with angiogenesis according to published data, and gene sets known to be positively associated with tumor-associated macrophage signatures (GO:0006955). Other tested gene sets included immune-related genes (GO:0006955), antitumor immune-response-related genes (GO:0002418), genes related to immune system processes (GO:0002376), and inflammatory genes associated with activated macrophages, all of which have been published. The association between OLFML3 expression and other genes was quantified by calculating the odds ratio (OR) using Fisher’s test, with expression stratified based on expression relative to the median value. Confidence intervals were obtained under the assumption that the log (OR) followed a normal distribution.

Forest plots and correlation plots were generated using the R library ‘ggplot2’ (https://www.springer.com/gp/book/9780387981413, accessed on 6 April 2019). All statistical analyses were performed using R software (version 3.5.3).

### 2.3. Animal Procedures

C57BL/6J immunocompetent mice and NSG immunodeficient mice were purchased from Janvier Labs (Le Genest Saint Isle, France) and the Jackson Laboratory, respectively. All animal procedures were performed in accordance with the guidelines set down by the Institutional Ethical Committee of Animal Care and the Swiss Cantonal Veterinary Office of Geneva and Fribourg (authorization numbers GE93/14 and GE150/17 for Geneva and 2016_06_FR for Fribourg). Male and female mice aged between 5 and 7 weeks were used.

Originally, the Olfml3 knockout allele was generated on a 129/Sv genetic background [15]. In the mONT3lacZ (Olfml3) knockout allele, the second codon of Olfml3 was replaced in frame with the attB1 site, following the LacZ sequence with an attached nuclear localization signal (nLacZ) [20]. Homozygous mutant mice were born normal and fertile. All Olfml3 knockout animals examined in this study were from the C57BL/6J genetic background [25]. Briefly, mice were backcrossed for 11 generations on a C57BL/6J background, resulting in 65% perinatal survival of Olfml3 KO animals without pathology [24]. Subsequent mice were obtained by F1 intercrosses, and 6-week-old mice were used for the MC38 tumor implantation studies.

### 2.4. Statistical Analysis and Expression of Results

Data are expressed as the mean ± SEM of the indicated number of experiments. Statistical analysis was performed using Prism software (GraphPad Software, San Diego, CA, USA). The variance between groups to be compared was estimated before statistical analysis. When comparing more than two groups, a nonparametric one-way ANOVA, followed by a Dunn’s multiple comparison test, was used. Significant differences between two groups were determined using the unpaired Student’s *t*-test or the Mann–Whitney test. A *p*-value ≤ 0.05 was considered significant. For the animal studies, the investigator was blinded to group allocation, and all mice were distributed randomly among the various groups.

## 3. Results

### 3.1. Expression of OLFML3 Is Increased in Human CRC and Is Associated with Shorter Relapse-Free Survival, Microsatellite Stability, and the Angiogenic CMS4 Subtype

We previously identified Olfml3 as a novel proangiogenic factor that promotes tumor growth in the LLC1 mouse lung tumor model [22]. However, the relevance of OLFML3 to human cancer is unreported. To address this, we first examined the expression of the OLFML3 protein in colorectal, kidney, lung, esophagus, prostate, and uterus carcinoma tissue sections and the corresponding healthy tissues using immunohistochemistry (IHC) or immunofluorescence (IF) staining. IF revealed strong OLFML3 expression in the tumor vasculature of colorectal and uterus cancers (Figure 1A,B, Appendix A), intermediate expression was observed in lung and prostate carcinomas (Appendix A), and low to undetectable expression was observed in the vessels of the kidney and esophageal cancers relative to healthy control tissues (Appendix A).

Based on these observations, we decided to further characterize OLFML3 expression in CRC. We analyzed the association between clinical-pathological characteristics and the expression of OLFML3 mRNA in three datasets (GSE39582, the Cancer Genome Atlas, and PETACC3) and observed a significant association between high expression of OLFML3 mRNA and shorter relapse-free survival (RFS) (*p* = 0.00079; Figure 1C). OLFML3 expression was significantly higher in stage 2–4 tumors than in stage 1 tumors (*p* < 0.04; Appendix A).

At the genomic level, CRC can be classified as chromosomal instable (CIN)/microsatellite stable (MSS) tumors or chromosomal stable/microsatellite instable (MSI) hypermutated tumors. At the transcriptomic level, CRC can be classified into four consensus molecular subtypes (CMS 1–4) [29]. Interestingly, stratification of MSS versus MSI tumors, along with CMS status, revealed elevated expression of OLFML3 in the MSS subtype relative to the MSI subtype (*p* = 0.1 × 10^−5^; Figure 1D) and the CMS4 subtype relative to the other three subtypes (Figure 1D). There was no difference between female and male patients (not shown).

Because the CMS4 subset is associated with angiogenesis and inflammation [29], we assessed the correlation between OLFML3 and angiogenic factor expression in a CRC RNA dataset (GO: 0001525) [30]. We observed a positive correlation between OLFML3 expression and the angiogenesis-associated factors ANGPT1, KDR, ANGPT2, FLT1, PECAM1, TEK, TIE1, and CDH5 (Figure 1E and Appendix A). Strikingly, VEGF-A did not significantly correlate with OLFML3 (Appendix A). Analysis of a second dataset (GO: 0045766) confirmed the correlations between OLFML3, ANGPT1, and TEK and revealed new significant correlations with CX3CR1, C3, FGF1, HGF, PRKD1, PTGIS, and RHOJ mRNA (Appendix A). Analysis of genes involved in the negative regulation of angiogenesis (GO: 0016525) revealed a strong negative correlation between OLFML3 and the antiangiogenic factors CREB3L1, EPHA2, E2F2, KLF4, EFNA3, and LIF (Appendix A). Finally, these analyses also revealed an unanticipated correlation between OLFML3 mRNA levels and those of podoplanin (PDPN) and FLT4, two genes implicated in lymphangiogenesis (Figure 1E, lower panels, and Appendix A).

These results demonstrate that OLFML3 is expressed in human tumors, particularly CRC, and colocalizes close to VE-cadherin-positive tumor-associated endothelial cells. In CRC patients, OLFML3 mRNA expression positively correlated with shorter RFS, the angiogenic CMS4 subtype, and transcripts encoding proangiogenic and lymphangiogenic factors, and negatively with antiangiogenic molecules, suggesting a role for OLFML3 in tumor angiogenesis and lymphangiogenesis.

### 3.2. OLFML3 Is Highly Expressed in CRC Patient-Derived Tumor Xenografts and Is Downregulated by Anti-VEGF-A/-VEGFR-2 Therapies in Mouse and Human CRC Transplant Models

Next, we evaluated OLFML3 expression in CRC patient-derived tumor xenografts (PDX) (21 primary tumors and 6 metastases of admitted tumor patients before the start of therapy, and transplanted into CB17-SCID mice (immunodeficient mice)). OLFML3 mRNA levels were increased in all primary and metastatic CRC samples grown as PDX in immunodeficient mice compared with human healthy colon tissue (dashed line in Figure 2A), and the increase was statistically significant (Figure 2B). The expression of OLFML3 in CRCs positively correlated with the expression levels of transcripts encoding molecules relevant to the vasculature, including PECAM1, VEGF-A, and TGF-β, the latter upregulating the expression of Olfml3 [31] (Figure 2C and Appendix A). Interestingly, we also found that TGF-β, CXCR4, and L1CAM correlate with OLFML3 in a CRC patient database (Appendix A). Interestingly, both CXCR4 and L1CAM are upregulated by TGF-β [32].

To determine whether OLFML3 expression in CRC is modulated by anti-VEGF therapy, we analyzed tumors from CRC patients treated (or not) with the anti-VEGF-A antibody bevacizumab (Avastin) [33]. NanoString nCounter mRNA expression array analysis revealed that bevacizumab treatment significantly reduced the expression of OLFML3 mRNA and the angiogenesis- and lymphangiogenesis-associated genes MCAM, NG2, and VEGF-C (Figure 2D).

To validate this observation in vivo, we grafted mouse MC38 CRC cells into C57BL6/J mice and human DLD1 CRC cells into NSG mice and treated them with anti-mouse Vegfr-2-blocking antibody (DC101) and bevacizumab, respectively. Tumor growth and final tumor size were significantly decreased in both models compared with controls (control IgG treatment) (Figure 2E and Appendix A). Bevacizumab treatment of DLD1-xenograft-bearing mice reduced the expression of mouse Olfml3 (host-derived) and human OLFML3 (tumor-derived, although not significantly) mRNA in tumor samples (Figure 2F). The expression of mouse Olfml3 mRNA correlated with the expression of the angiogenesis-associated genes Pecam1, Mcam, Vegfr-2, Vegfr-1, and Vegf-a (Figure 2G), consistent with the above ex vivo results. The correlation between human OLFML3 and VEGF-A was not statistically significant (Appendix A).

These findings suggest that the inhibition of the VEGF/VEGFR signaling pathway may regulate the expression of human and mouse OLFML3 within the tumor stroma of human and mouse CRCs, respectively. Further analysis of this experiment revealed that the expression of OLFML3 in bevacizumab-treated tumor grafts correlated with the expression of the angiogenesis-related genes described above.

### 3.3. Treatment of Mice with Anti-OLFML3 Recombinant Antibodies or Olfml3 Gene Deletion Inhibits Colorectal Tumor Growth and Angiogenesis

Next, we tested the potential therapeutic effects of two novel recombinants (rec) anti-OLFML3 antibodies (Abs), rec9F8 and rec46A9, which were generated recently based on rat OLFML3 monoclonal antibodies (mAbs) produced in our laboratory (Appendix A). The recombinant antibodies retained their original affinity and recognized two different epitopes within peptide B of OLFML3 (Appendix A). When compared with our previous polyclonal rabbit antibodies, the new reagents inhibited mouse LLC1 lung tumor growth (Appendix A) and could be relevant for therapeutic application [22].

MC38 CRC cells were grafted subcutaneously into immunocompetent C57BL6/J mice, and antibody treatment started 5 days later (Figure 3A). Both of the rec-anti-OLFML3 antibodies decreased MC38 tumor growth significantly relative to control antibody treatment (Figure 3B). To assess the effect of rec46A9 and rec9F8 on MC38 tumor vascularization, we performed Pecam1 IHC staining (Figure 3C) and determined the frequency of vascular and lymphatic endothelial cells in tumor cells suspensions using CD45/Pecam1/Pdpn staining of endothelial cells by flow cytometry (Figure 3D). We observed a significant decrease in Pecam1-positive cells immunostaining in tumors from rec9F8- and rec46A9-treated mice relative to IgG control treatment (Figure 3C) and a lower fraction of vascular (CD45^−^/Pecam1^+^/Pdpn^−^) and lymphatic (CD45^−^/Pecam1^+^/Pdpn^+^) endothelial cells in tumors from rec9F8-treated mice (Figure 3D). Additionally, the Pecam1+ structures in the IHC show a different pattern. Indeed, Pecam1^+^ cells in IgG-treated tumors are more elongated than in anti-OLFML3-treated tumors, where they are smaller and round, consistent with blood vessel regression.

To explore the role of host Olfml3 during tumor growth and vascularization, we subcutaneously inoculated wild-type (WT) and Olfml3 knockout mice [25] with MC38 CRC cells. Deletion of Olfml3 led to a significant decrease in MC38 tumor growth and volume relative to those in WT animals (Figure 3E). We observed a significant decrease in Pecam1^+^ immunostaining in tumors from knockout mice relative to WT mice (Figure 3F) and a decrease in the frequency of vascular and lymphatic endothelial cells in anti-OLFML3 antibody-treated mice using flow cytometry analysis (Figure 3G).

These findings demonstrate that treatment of tumor-bearing mice with recombinant anti-OLFML3 mAbs and Olfml3 host gene deletion inhibits lymphangiogenesis and CRC tumor growth.

### 3.4. Targeting OLFML3 Blocks Blood Vessel Formation and Pericyte Recruitment in Tumors

Previously, we reported that OLFML3 promotes PDGF-dependent pericyte recruitment during embryonic blood vessel formation [25]. To determine whether inhibiting OLFML3 also prevents the recruitment of pericytes to tumor blood vessels, we treated LLC1-bearing mice with anti-OLFML3 antibodies and then analyzed tumor vascularization and pericyte coverage by double IF staining of tumor sections for Pecam1 (vascular endothelial cells) and α-Sma (pericytes). Treatment with anti-OLFML3 antibodies reduced both vascularization and pericyte coverage in LLC1 tumors when compared with controls (Figure 4A).

As pericytes control the maturation and integrity of vascular endothelium, the antiangiogenic (and antitumor) effects of OLFML3 targeting might also involve pericytes. To test this at the cellular level, we established an in vitro endothelial–pericyte coculture assay mimicking the recruitment of pericytes to endothelial cells. Human pericytes (labeled with a green cell tracker) were cocultured with human umbilical vein endothelial cells (HUVECs; labeled with a red cell tracker) plated on Matrigel (Figure 4B). Over a 7-h time course, pericytes aligned with HUVECs and formed a capillary-like network (Figure 4B). In the presence of 9F8 and rec9F8, the complexity of the branching network was reduced (Figure 4B and Appendix A, micrographs). Differences were confirmed by the measurement of the total branch length of the capillary-like structures (Figure 4B and Appendix A, graphs). A time-course study demonstrated that the 9F8 antibody was more efficient than 46A9 at inhibiting endothelial branching, reaching a plateau after 3–5 h and with a significantly lower total branching length at 9 h (Figure 4B, graph). Importantly, the rec9F8 antibody retained the angiogenesis-inhibiting capacity of the original mAb (Appendix A, graph).

These results point to pericytes as an additional putative cellular target of OLFML3 blockade. In line with this hypothesis, qPCR analysis confirmed that both primary mouse and human brain pericytes express Olfml3 and OLFML3, respectively (Appendix A). We conclude that tumor-associated endothelial cells and pericytes express Olfml3.

Above we found that inhibiting VEGF-A triggers the compensatory expression of proangiogenic factors, including other VEGF family members [34]. Therefore, we also monitored the effect of tumor treatment with anti-OLFML3 antibodies on the expression of human and mouse angiogenesis-associated transcripts in human DLD1 CRC tumors treated with rec9F8, or bevacizumab. Analysis of NanoString nCounter mRNA arrays (Appendix A) revealed that treatment of DLD1-bearing mice with rec9F8 reduced the expression of mRNA encoding the mouse stromal cell-derived factors Pdgfr-β, Vegf-c, Vegfr-1, Vegfr-2, and Olfml3, whereas bevacizumab decreased the levels of Pdgfr-β, Vegfr-1, and Vegfr-2, but not Olfml3 mRNA (Figure 4C). Importantly, while bevacizumab increased the expression of VEGF-A mRNA, consistent with a compensatory reaction, rec9F8 had no such effect (Appendix A). The ELISA of plasma showed that anti-mouse VEGFR-2 inhibition with DC101 mAb increased the concentrations of both Vegf-a and Vegf-c (Figure 4D,E). The anti-OLFML3 antibodies did not increase the plasma concentrations of Vegf-a, while they decreased the concentration of Vegf-c (Figure 4E). This decrease in Vegf-c levels is consistent with the reduced number of lymphatic endothelial cells in the tumors of animals treated with anti-OLFML3 mAbs (Figure 3D). Accordingly, analysis of human mRNA expression datasets revealed that the expression of OLFML3 correlates with that of VEGF-C in human samples (Figure 4F).

These findings demonstrate that antibody targeting of OLFML3 on endothelial cells and pericytes attenuates the association of pericytes with cocultured endothelial cells in vitro and angiogenesis and pericyte recruitment in tumors in vivo. Anti-Olfml3 antibody treatment does not cause a compensatory increase in proangiogenic growth factors.

### 3.5. Both OLFML3 Recombinant Antibodies and Olfml3 Gene Deletion Target TAMs

Angiogenic factors promote myeloid cell recruitment and function, and antiangiogenic therapy increases tumor infiltration by immune cells [35]. Targeting the vascular endothelium and pericytes also affects leukocyte infiltration. Macrophage subpopulations have pro- or antitumor effects and modulate tumor angiogenesis [36]. Therefore, we assessed the effects of the rec9F8 antibody and Olfml3 gene deletion on immune cell infiltration in MC38 tumors. Macrophages (CD45^+^/CD11b^+^/F4-80^+^) and a subpopulation of TAMs (CD45^+^/CD68^+^/CD11b^+^/F4-80^+^), determined by flow cytometry (Appendix A), were significantly reduced in tumors of rec98F-treated and Olfml3-deficient mice (Figure 5A). We did not observe a significant reduction in the percentage of proinflammatory TAMs (MHCII^+^/Ly6C^+^) (Appendix A). To test whether the reduced number of TAMs was responsible for the reduced tumor angiogenesis after therapy with rec9F8 or in Olfml3 knockout mice, we experimentally depleted macrophages by treating tumor-bearing mice with clodronate liposomes. This did not affect tumor angiogenesis and lymphangiogenesis, whereas rec9F8 antibody treatment reduced angiogenesis in clodronate-treated mice (Figure 5B). Growth curves and ex vivo measurement of tumor volume revealed that both clodronate liposomes and rec9F8 treatment inhibited tumor growth relative to the respective controls (Figure 5C, upper panels), while analysis of tumor cell suspensions by flow cytometry confirmed the reduced TAM counts in clodronate- and rec9F8-antibody-treated mice (Figure 5C, lower panels, Appendix A). Again, no significant difference in the percentage of proinflammatory TAMs was observed (Appendix A). Combined treatment with clodronate plus the rec9F8 antibody further increased the antiangiogenic and antitumor growth effects (Figure 5B,C). Thus, we conclude that the TAMs inhibited by the rec9F8 antibody or clodronate liposomes are of the tumor-promoting type.

To assess whether OLFML3 expression correlates with the number of macrophage-associated factors in human tumors, we analyzed the same CRC patient datasets (GO:0006955) and found a strong correlation between OLFML3 and the expression of the oncostatin M receptor (OSMR), CCL2 and CCL7 chemokines, and their respective receptors CXCR5 and CXCR2 (Appendix A). Both the rec9F8 antibody and bevacizumab significantly decreased the levels of the human-tumor-derived angiogenic factor PLGF (Appendix A). Strikingly, anti-OLFML3 mAb treatment decreased the plasma levels of mouse Plgf in human DLD1 and mouse MC38 tumor models, whereas DC101 treatment had the opposite effect (Figure 5D). Additionally, PLGF, TEK, and the CSF1 receptor (CSF1-R), which are involved in the recruitment, accumulation of monocytes, and their differentiation into macrophages [37], correlated with OLFML3 expression (Figure 5E).

The CSF1/CSF-R axis promotes macrophage proliferation, differentiation, and survival [38]; therefore, we set out to collect additional evidence for macrophage involvement in the antitumor effects of the anti-OLFML3 antibody. We treated MC38 tumor-bearing mice with rec9F8 or an anti-CSF1-R blocking antibody and found that both decreased MC38 tumor growth, as previously reported [39] (Figure 5F). However, combined treatment with the two antibodies had no additive effect on MC38 tumor growth (Figure 5F), suggesting that OLFML3 may act along the CSF1/CSF1-R axis.

To confirm that recombinant anti-OLFML3 antibodies did not induce tumor cell death via ADCC or complement, we generated a glycosylation-deficient rec46A9 variant (rec46A9M) bearing a mutation in the Fc region at site N297 (N297Q) that prevents mAb recognition of Fc receptors by complement effectors. Like rec46A9, the rec46A9M antibody was still able to inhibit MC38 tumor growth and tumor angiogenesis (Appendix A), excluding the possibility of ADCC or complement-mediated killing as the mechanism of action.

These findings demonstrate that OLFML3 expression correlates with tumor-associated macrophage signatures in CRC patients and that anti-OLFML3 therapy impairs the recruitment of tumor-promoting TAMs to CRC tumors. Thus, TAMs are additional targets of anti-OLFML3 treatment.

### 3.6. Anti-OLFML3 Antibody Treatment Increases the Anti-Tumor Effect of Anti-PD-1 Immunotherapy

Immune checkpoint inhibition with anti-PD-1 blocking antibodies hinders MC38 tumor development in mice by reactivating immune responses [40,41]. Emerging evidence indicates that antiangiogenic therapies modulate immune cell recruitment into the tumor microenvironment and increases the antitumor activity of immune checkpoint inhibitors [42,43]. In light of these observations, we sought to determine whether rec9F8 might enhance the efficacy of anti-PD-1 treatment in the MC38 model. Combined treatment with rec9F8 and anti-PD-1 antibodies increased the efficacy of anti-PD-1 monotherapy by more than 50% (Figure 6A). Interestingly, we observed that one of the eight mice treated with anti-PD-1 antibody did not develop a primary tumor, while, remarkably, three of the eight mice did not develop a primary tumor with the combination of anti-PD-1 and anti-OLFML3. In addition, it did not further affect the recruitment of macrophages, TAMs, CD4^+^, or CD8^+^ T lymphocytes compared with anti-PD-1 monotherapy alone (Figure 6B,C), while it increased the recruitment of proinflammatory macrophages, B lymphocytes (CD45^+^/CD3^−^/B220^+^), and NKT cells (Figure 6D,E). We observed no changes in the frequency of intratumoral CD4^+^ and CD8^+^ T lymphocytes in anti-OLFML3-antibody-treated or Olfml3-deficient mice (Appendix A), while NKT cell numbers were significantly increased in MC38 tumors under these two conditions. Moreover, this increase was not abrogated by clodronate treatment (Appendix A).

Analysis of the CRC patient dataset revealed that the expression of OLFML3 mRNA significantly correlated with the expression of mRNA encoding molecules specific for activated immune and inflammatory cells, immune checkpoint inhibitors PD-L1, PD-1, and CTLA4, and with the expression of HAVCR2, a molecule expressed by activated IFN-γ-producing immune cells (Figure 6F,G and Appendix A).

These findings demonstrate that anti-OLFML3-blocking antibodies strengthen the effect of anti-PD-1 therapy on MC38 tumor growth independently of ADCC or complement toxicity. This effect was associated with an increased presence of intratumoral NKT cells, B cells, and proinflammatory macrophages and with a reduced percentage of immunosuppressive TAMs. In human CRC tumors, the expression of OLFML3 strongly correlated with markers specific for activated immune cells.

## 4. Discussion

In this study, we used bioinformatics analysis of human datasets, immunostaining and mRNA expression analyses of tumor samples, human PDX models, and human/mouse tumor graft models to demonstrate that OLFML3 is overexpressed in multiple carcinomas relative to the corresponding healthy tissues. Detailed analyses revealed that OLFML3 expression in MSS CRC is higher than in MSI tumors, higher in grade 2–4 than in grade 1 tumors, and higher in the CMS4 subtype. Interestingly, the CMS4 subtype is characterized by stromal remodeling, TGF-β signaling, invasion, tumor angiogenesis, and low immune cell infiltration (in particular by T, B, and NK cells) [29]. Immunohistological analyses revealed that OLFML3 colocalizes with VE-cadherin-positive (vascular and lymphatic) endothelial cells and pericytes. Analysis of primary and passaged tumors from CRC patients revealed that OLFML3 expression significantly correlates with the expression of proangiogenic factors of the VEGF family and lymphangiogenesis markers, such as VEGF-C, PDPN, and FLT4. The new anti-OLFML3 antibodies developed for therapeutic purposes inhibit tumor growth, tumor angiogenesis, lymphangiogenesis, pericyte coverage, and the expression of lymphangiogenic factors.

Anti-VEGF-A treatment impinges on tumor vessels without altering their pericyte coverage [44]. Leaving pericytes untouched allows rapid recovery of tumor vessels and subsequent resistance to anti-VEGF therapy [41]. In addition, inhibiting VEGF-A triggers compensatory, proangiogenic reactions contributing to therapy resistance [34,42]. Here, we confirmed a compensatory increase in the expression of a VEGF family member mRNA and protein following the treatment of tumor-bearing mice with anti-VEGF-A and anti-Vegfr-2 antibodies. Maintenance of pericytes and their associated vascular basal membranes, along with the elevated secretion of proangiogenic factors, leads to rapid and physiologically relevant regrowth of tumor vessels after cessation of anti-VEGF-A therapy, thereby contributing to therapeutic failure [43,44,45,46,47]. Targeting OLFML3 with monoclonal and recombinant antibodies inhibits tumor angiogenesis and tumor growth and has two notable advantages over anti-VEGF-A therapy: it disrupts the pericyte coverage of endothelial cells, targets angiogenic endothelial cells themselves, and prevents a compensatory proangiogenic response. The results obtained with anti-OLFML3 were confirmed in Olfml3 knockout mice.

Taken together, our observations suggest that inhibiting OLFML3 stimulates antitumor immunity, possibly via effects on tumor blood and lymphatic vessels, and inflammatory cells. When stimulated by VEGF-A, tumor-associated blood vessels acquire molecular and functional phenotypes that limit T cell trafficking [48,49]. Extravasation of T cells is a multistep process that includes binding to the endothelium via vascular adhesion molecules such as VCAM-1 and ICAM-1 expressed on the luminal cell surface. VEGF-A downregulates the expression of these adhesion molecules on tumor blood vessels, thereby limiting T cell adhesion and extravasation [50]. Furthermore, tumor-associated endothelial cells express proteins such as PD-1, which abrogate the activation and function of T cells newly recruited to tumors [51,52]. Another line of experimental evidence shows that VEGF-A suppresses antitumor immunity by affecting other immune cells. Indeed, in association with immune checkpoint blockade, targeting of VEGF-A improves NK cell responses to chemotherapy and improves T-cell-mediated antitumor immunity [53,54]. We found that in colorectal tumors, recombinant anti-OLFML3 antibody treatment decreases the abundance of mature macrophages and TAMs while increasing the recruitment of NK-like T cells. Therefore, macrophages may be an important target of anti-OLFML3 therapy; this idea is supported by the strong correlation between OLFML3 expression and macrophage-associated molecules, such as PLGF, CSF1R, TEK, and HGF. Experiments using a combination of anti-OLFML3 and anti-CSF1-R antibodies or clodronate liposomes showed an additional therapeutic effect only with clodronate relative to anti-OLFML3 antibody alone, supporting the notion that both drugs act, at least in part, on a common target. These effects are consistent with the observed decrease in Plgf expression after anti-OLFML3 treatment. PLGF attracts and activates monocytes to produce proangiogenic and lymphangiogenic molecules in the tumor microenvironment [55]. PLGF also inhibits differentiation of antigen-presenting dendritic cells and contributes to tumor growth, neovascularization, and lymphangiogenesis by increasing the recruitment of FLT1-expressing hematopoietic progenitor cells and monocytes [56,57]. These data suggest that the antitumor effects of OLFML3 inhibition involved events mediated by the PLGF and CSF1 pathways.

We extensively described that several important tumor-growth-promoting cell types are affected by anti-OLFML3 antibody treatment (blood and lymphatic endothelial cells, pericytes, and TAMs) that are altogether linked during tumor development, blood vessel formation, and immune cell infiltration (Figure 7). It seems plausible that these multiple combined effects concur to suppress tumor growth. However, we cannot exclude a direct effect on tumor cells (i.e., apoptosis, proliferation, senescence) or other cell types of the tumor microenvironment (i.e., fibroblasts). This may be addressed in future studies.

Clinical and experimental evidence indicates that antiangiogenic drugs profoundly modulate the composition of immune cells in the tumor microenvironment, thereby affecting antitumor immunity [58]. For example, bevacizumab therapy promotes infiltration by CD4^+^ and CD8^+^ T cells and stimulates PD-L1 expression in carcinomas [59]. In our models, anti-OLFML3 antibodies promoted tumor homing of NK-like T cells but did not increase the recruitment of T lymphocytes. Although the mechanism of OLFML3-mediated NK-like T cell recruitment is not yet clear and is outside of the scope of this study, NK cells promote tumor cell apoptosis and secrete the immune-stimulatory cytokine IFN-γ [60]. Notably, the expression of OLFML3 correlates with the expression of immune checkpoint inhibitors, including the PD-L1/PD-1 mRNA, in human tumor samples. This led us to study the effect of anti-OLFML3 antibodies on anti-PD-1 antibody treatment. The combination of anti-OLFML3 and anti-PD-1 antibodies increases tumor infiltration by inflammatory myeloid cells, B cells, T lymphocytes, and NKT cells, and inhibits tumor growth more effectively than either agent used alone. Such combinations of angiogenesis-targeting agents and checkpoint inhibitors have been recommended for the treatment of the poorly immunogenic CMS4 CRC subtype [12,13,14,15,16,17,18,19,20,21,22,23,24,25,26,29,30,31,32,33,34,35,36,37,38,39,40,41,42,43,44,45,46,47,48,49,50,51,52,53,54,55,56,57,58,59,60,61]. Future studies are needed to determine how to best exploit the therapeutic potential of combined OLFML3 inhibition and treatment with immune checkpoint inhibitors. Because breast cancers are poorly, or at best moderately, responsive to anti-VEGF-A therapy and immune checkpoint blockade [62], it would be interesting to study the effect of inhibiting OLFML3 in breast cancer models. This may be of particular interest for preventing the metastatic progression of triple-negative breast cancers infiltrated by immune cells [63,64].

## 5. Conclusions

In conclusion, our findings demonstrate the therapeutic benefits of antibody-mediated targeting of OLFML3 and provide important insight into the underlying mechanisms, which involve the inhibition of tumor angiogenesis, lymphangiogenesis, pericyte coverage, and immune cell infiltration (Figure 7). The results suggest that anti-OLFML3 therapy is an attractive standalone therapy or an effective combination therapy with immune checkpoint inhibitors.

## Figures and Tables

**Figure 1 cancers-13-04625-f001:**
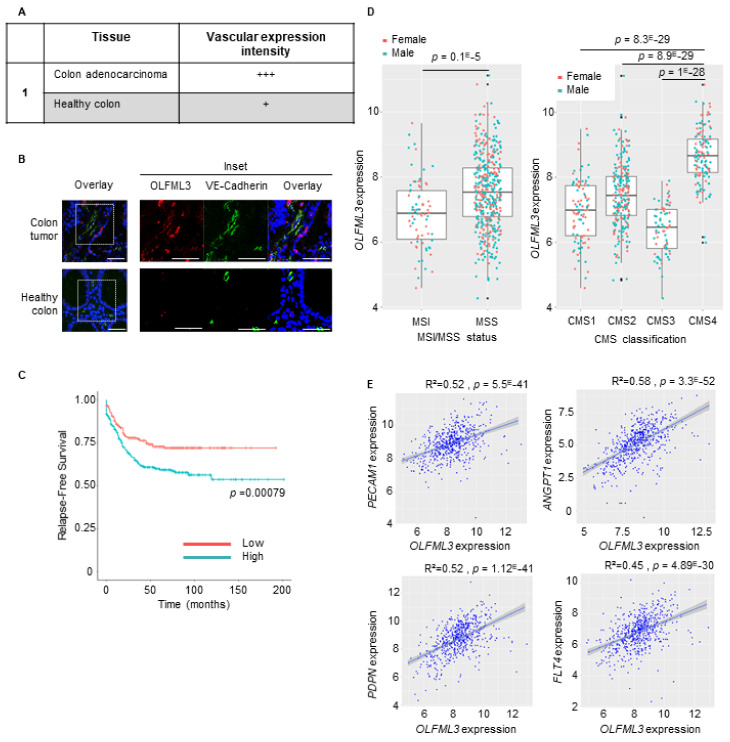
Expression of OLFML3 in colorectal cancer (CRC) correlates with shorter relapse-free survival, with microsatellite stable status (MMS) and consensus molecular subtypes 4 (CMS4), and with the expression of angiogenic factors. (**A**) Vascular expression intensity of OLFML3 was ranked as follows: none (−), low (+), moderate (++), or strong (+++). (**B**) Representative confocal images of VE-cadherin+ (green)- and OLFML3+ (red)-stained tumor vasculature in sections of CRC and healthy colon tissue. Note the difference in abundance of vascular VE-cadherin+ and perivascular cells in OLFML3-expressing tumors relative to that in healthy colon tissue. DAPI (blue), nuclear counterstain (overlays). Scale bars = 50 μm. The right panel is a zoom image of the frame of the left panel. (**C**) Kaplan–Meier survival analysis of the index of OLFML3 mRNA expression (low (red) vs. high (blue)) relative to the median value in CRC patient tumor samples. ((**D**), left) Expression of OLFML3 mRNA in CRC patient tumor samples according to microsatellite instable/microsatellite stable status (MSI/MSS). ((**D**), right) Expression of OLFML3 mRNA in four consensus molecular subtypes (CMS 1–4) from CRC patients. (**E**) Scatter plots with linear regression lines (blue) showing the correlation between levels of OLFML3 mRNA and that of angiogenic markers (PECAM1, ANGPT1, PDPN, FLT4) in CRC samples and normal colon tissue. Levels of OLFML3 mRNA in different subgroups of patients were compared using a *t*-test with equal variance. The correlation between the expression of OLFML3 and that of genes of interest was calculated using Pearson’s and Spearman’s rank correlation analyses.

**Figure 2 cancers-13-04625-f002:**
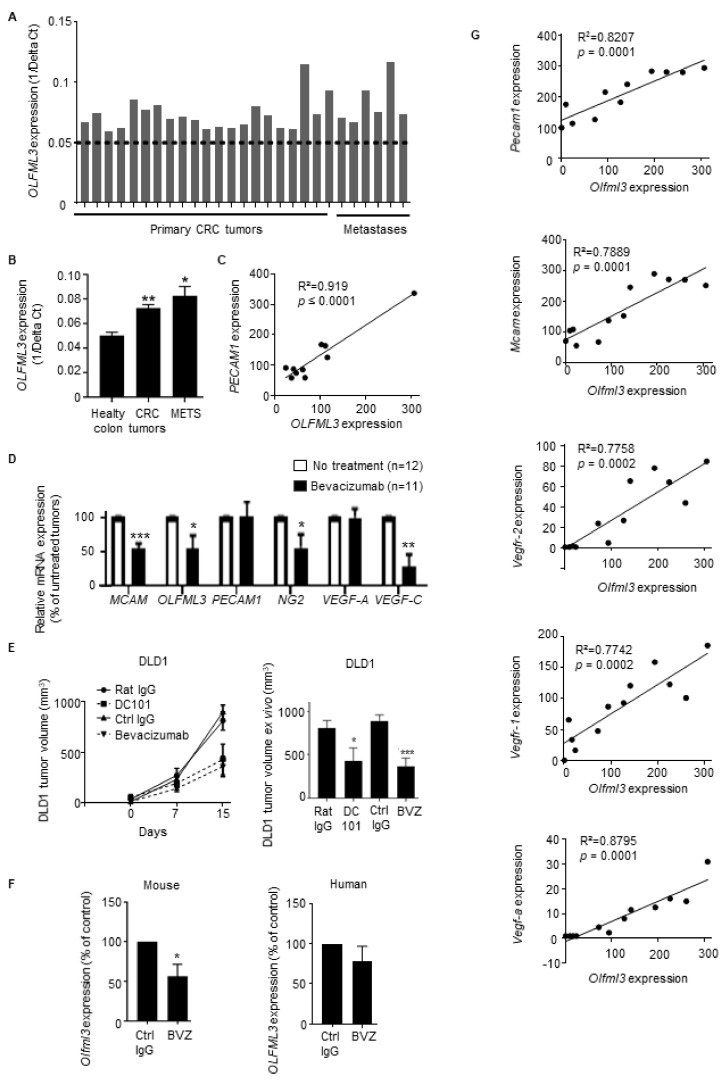
Expression of OLFML3 mRNA in primary CRC tissues, patient-derived xenografts (PDXs), and tumor transplantation models correlates with that of angiogenic factors and decreases after anti-VEGF therapy. (**A**) Relative expression of OLFML3 mRNA in PDX samples derived from primary CRC tumors (*n* = 21) or CRC metastatic lesions (*n* = 6) relative to the average expression level in healthy colon tissue (dotted line), as determined by qRT-PCR. The average CT value was calculated for both OLFML3 and the housekeeping gene EEF1; delta CT (CT, OLFML3-CT, EEF1) was determined. (**B**) Relative mean OLFML3 mRNA levels in the tissues described in (**A**). (**C**) Scatter plots with linear regression lines (black) showing the correlation between the expression of Olfml3 and PECAM1 mRNA in PDX samples (*n* = 10), as determined by NanoString mRNA expression arrays. (**D**) Relative expression of mRNA encoding OLFML3 and angiogenic factors in tumor biopsies from CRC patients treated (*n* = 11) or not (*n* = 12) with bevacizumab, as determined by NanoString mRNA expression arrays. Results are normalized against values in nontreated patients. Bars represent the means ± SEM. (**E**) Growth of subcutaneous DLD1 xenografts (left) and size of excised tumors at the time of sacrifice (right) in immunodeficient NSG mice treated with an antihuman VEGF-A blocking mAb (bevacizumab), DC101, or the corresponding control IgG mAbs (*n* = 6 mice per group). (**F**) Relative expression of mRNA encoding mouse Olfml3 (left) and human OLFML3 (right) in human DLD1 CRC tumor samples from NSG mice treated with either bevacizumab (*n* = 6) or control IgG (*n* = 6). Results are normalized against the values in nontreated mice. Bars represent the mean ± SEM; * *p* < 0.05. (**G**) Scatter plot with linear regression lines (black) showing the correlation between NanoString mRNA expression profiles of Olfml3 and levels of transcripts encoding mouse angiogenic factors (Pecam1, Mcam, Vegfr-2, Vegfr-1, and Vegf-a) in DLD1 tumor extracts (*n* = 12), as determined by Pearson’s and Spearman’s rank correlation analyses. Differences between the groups were evaluated using the Mann–Whitney or unpaired Student’s *t*-test. Bars represent the mean ± SEM; * *p* < 0.05; ** *p* < 0.01; *** *p* < 0.01.

**Figure 3 cancers-13-04625-f003:**
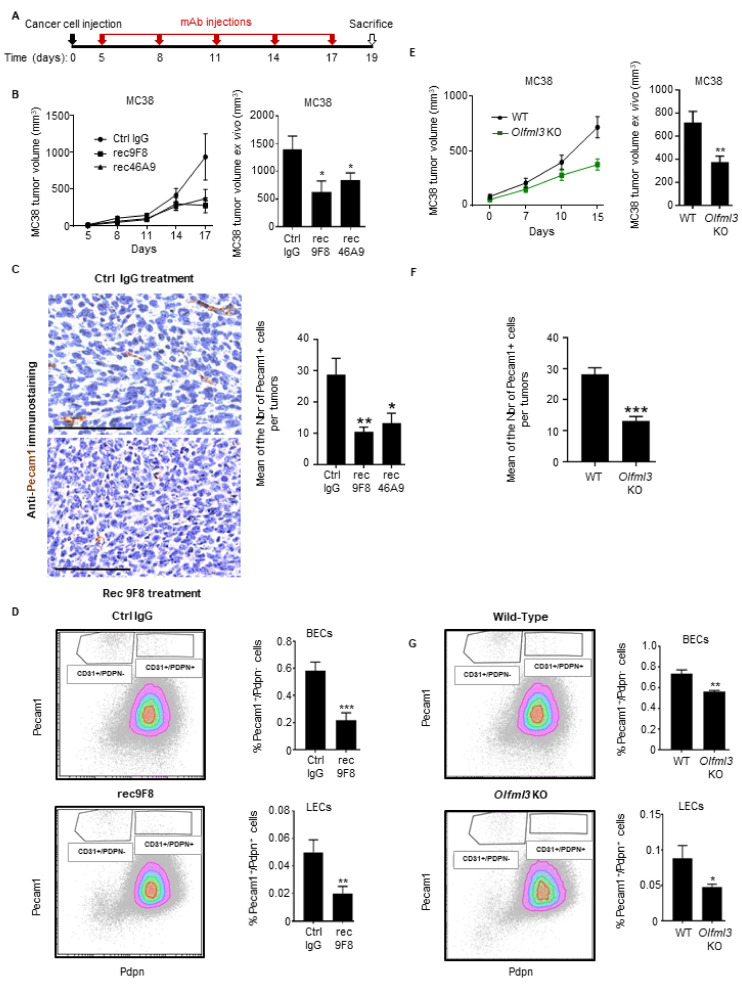
Expression of OLFML3 in inhibitory effects of anti-OLFML3 mAbs and Olfml3 gene deletion on CRC tumor growth and vascularization. (**A**) C57Bl6/J mice received a subcutaneous implant of MC38 tumor cells, followed by treatment with anti-OLFML3 or isotype control mAbs on days 0, 5, 8, 11, 14, and 17. (**B**) The volume of MC38 tumors in mice treated with rec9F8 (*n* = 7), rec46A9 (*n* = 7), or IgG2a isotype control mAb (Ctrl IgG, *n* = 8) was measured every 2–4 days (starting on day 3) using a caliper (left) or ex vivo after sacrifice on day 19 (right). Bars (right) represent the mean ± SEM; * *p* < 0.05. (**C**) Right, representative images (scale bar, 100 µm), and left, quantification of Pecam1-positive events after Pecam1 immunostaining of tumor sections from (**B**). Results represent mean values ± SEM; * *p* < 0.05; ** *p* < 0.01. (**D**) Gating strategy used for flow cytometry analysis of CD45^−^/CD31^+^/PDPN^−^ vascular endothelial cells and CD45^−^/CD31^+^/PDPN^+^ lymphatic endothelial cells in single-cell suspensions derived from MC38 tumors from mice treated with rec9F8 or control IgG antibodies (left). Quantification of vascular CD45^−^/CD31^+^/PDPN^−^ (middle) and lymphatic CD45^−^/CD31^+^/PDPN^+^ (right) endothelial cells. (**E**) Growth curve (left) and ex vivo measurement of the volume (right) of MC38 tumors growing in wild-type (WT) (*n* = 9) or Olfml3 knockout mice (*n* = 14). Bars (right) represent the mean ± SEM; * *p* < 0.01. (**F**) Quantification of Pecam1-positive events after Pecam1 immunostaining of the tumor sections (**E**). Results represent the mean value ± SEM; * *p* < 0.05; ** *p* < 0.01. (**G**) Gating strategy used for flow cytometry analysis of CD45^−^/CD31^+^/PDPN^−^ vascular endothelial cells and CD45^−^/CD31^+^/PDPN^+^ lymphatic endothelial cells in MC38 tumor-derived single-cell suspensions from WT and Olfml3 knockout mice (left). Quantification of vascular CD45^−^/CD31^+^/PDPN^−^ (middle) and lymphatic CD45^−^/CD31^+^/PDPN^+^ (right) endothelial cells. Results represent the mean value ± SEM; * *p* < 0.05; ** *p* < 0.01; *** *p* < 0.01.

**Figure 4 cancers-13-04625-f004:**
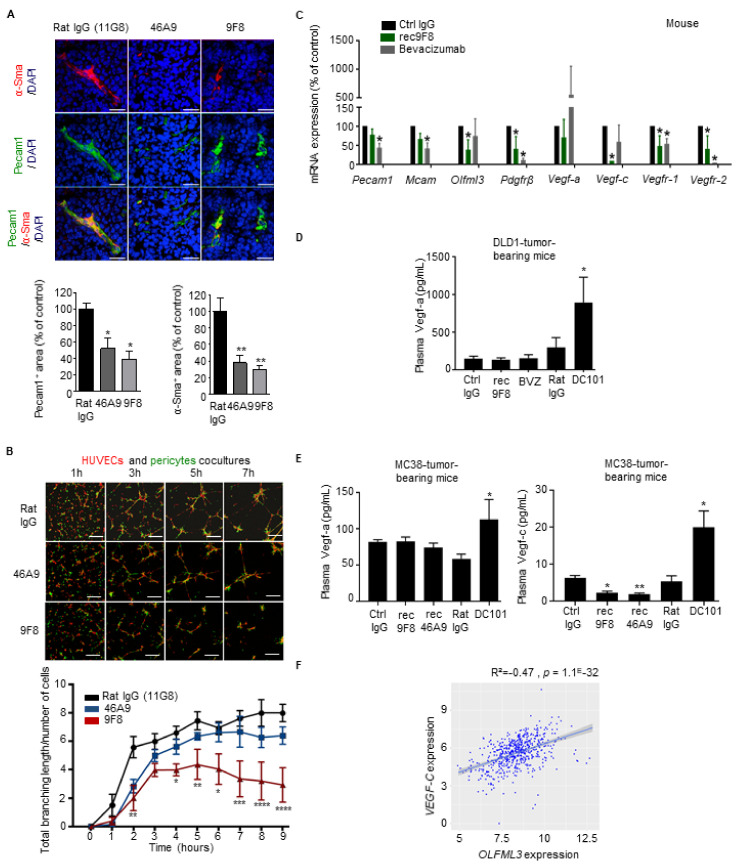
Anti-OLFML3 antibody treatment decreases vascularization, pericyte coverage, and expression of angiogenesis-associated factors in colorectal and lung tumors. (**A**) Vascularization of LLC1 lung tumors. Upper panel: representative confocal images comparing α-Sma+ pericytes (red), Pecam1+ endothelium (green), and Pecam1+/α-SMA+ double staining (green and red) in LLC1 tumors grown in mice treated with isotype control rat IgG (11G8) or the anti-Olfml3 antibodies 46A9 or 9F8. DAPI (blue), nuclear counterstain. Bars, 100 µm. Lower panels: quantification of vessel density, expressed as the ratio of the total pixel count for Pecam1 to that for DAPI, normalized against control rat IgG (left). Quantification of pericyte density expressed as the ratio of total pixel count for α-Sma to that for DAPI, normalized against control rat IgG (right). Ten individual images of tumors, taken in three planes, were analyzed (five mice per group). Error bars represent ± SEM; * *p* < 0.05; ** *p* < 0.01. (**B**) Coculture of HUVECs with human brain pericytes. Upper panel: representative confocal images taken at the indicated times after seeding of cocultures of HUVECs (red) and human brain pericytes (green) in 3D Matrigel in the presence of 9F8, 46A9, or rat control mAbs. Lower panel: quantification of total branch length of capillary-like networks. Results represent the mean value ± SEM (three experiments, each group analyzed in triplicate); * *p* < 0.05; ** *p* < 0.01;*** *p* < 0.001; **** *p* < 0.0001. (**C**) Expression profile of mouse mRNA expression using NanoString arrays of DLD1 CRC tumors. Angiogenic factors in DLD1 tumors from mice treated with rec9F8, bevacizumab, or control IgG mAbs. Results are normalized against the values for control IgG-treated tumors. Columns represent the mean ± SEM (three experiments); * *p* < 0.05. (**D**,**E**) ELISA measurement of angiogenesis-associated factors in plasma of tumor-bearing mice. (**D**) Vegf-a in mice bearing DLD1 human CRC tumors treated with rec9F8, bevacizumab, DC101, or the corresponding control IgG antibodies (Ctrl IgG, Rat IgG). (**E**) Quantification of Vegf-a or Vegf-c in mice bearing MC38 mouse CRC tumors treated with rec9F8, rec46A9, DC101, or control IgG antibodies. (**F**) Scatter plots with linear regression lines (blue) showing the correlation among the NanoString mRNA expression profiles of OLFML3 and VEGF-C mRNA levels in tumor samples from CRC patients, calculated using Pearson’s and Spearman’s rank correlation analyses. Differences between groups were determined using an unpaired Student’s *t*-test.

**Figure 5 cancers-13-04625-f005:**
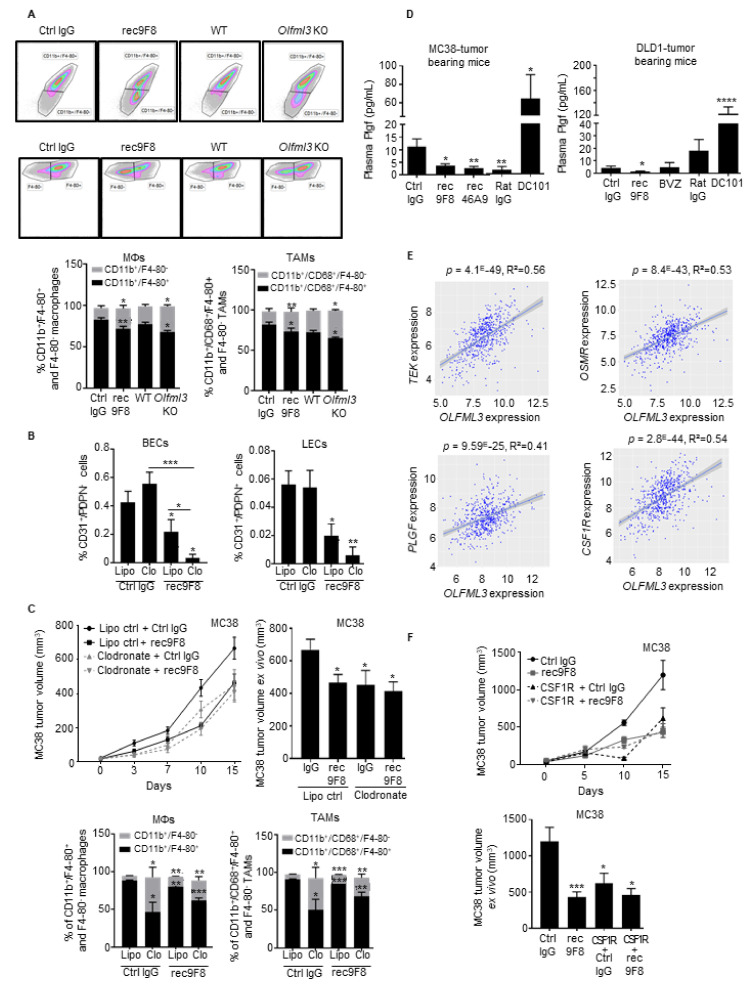
Anti-OLFML3 antibodies or deletion of Olfml3 from mice reduces the percentage of tumor-associated macrophages (TAMs) in MC38 tumors. (**A**) Upper panel: flow cytometry gating strategy for immature and mature macrophages (upper plot) and TAMs (lower plot) in single-cell suspensions of MC38 tumors in wild-type (WT) mice treated with control or rec9F8 antibodies, or in Olfml3 WT or KO mice. Lower panel: quantification of immature (CD11b^+^/F4-80^−^) and mature (CD11b^+^/F4-80^+^) macrophages (left) and immature (CD11b^+^/CD68^+^/F4-80^−^) and mature (CD11b^+^/CD68^+^/F4-80^+^) TAMs (right). Columns represent the percentage of cells, and error bars represent the mean ± SEM; * *p* < 0.01; ** *p* < 0.01. (**B**) Quantification of blood vascular (BECs; CD31^+^/PDPN^−^) and lymphatic (LECs; CD31^+^/PDPN^+^) endothelial cells in MC38 tumors isolated from mice treated with clodronate-loaded liposomes, rec9F8 mAbs, and their respective controls. Columns represent the mean ± SEM; bars (right), * *p* < 0.05; ** *p* < 0.01; *** *p* < 0.001. (**C**) Growth curves (left) and size of ex vivo MC38 tumors at the end of the experiment (right) from mice treated with rec9F8 mAb alone or in combination with clodronate-loaded liposomes, control liposomes, or control mAbs (*n* = 8–11 per group). Lower panels: Quantification of immature (CD11b^+^/F4-80^−^) and mature (CD11b^+^/F4-80^+^) macrophages (left) and immature (CD11b^+^/CD68^+^/F4-80^−^) and mature (CD11b^+^/CD68^+^/F4-80^+^) TAMs (right) in MC38 tumors isolated from mice treated with clodronate-loaded liposomes, rec9F8, and their respective controls. Bars (right) represent the mean ± SEM; * *p* < 0.05; ** *p* < 0.01; *** *p* < 0.001. (**D**) Quantification of Plgf plasma levels in mice bearing DLD1 tumors treated with rec9F8, bevacizumab, DC101, or the corresponding IgG control mAbs (Ctrl IgG, Rat IgG) (right), and in mice harboring MC38 tumors treated with rec9F8, rec46A9, DC101, or the corresponding control IgG antibodies (left). Bars represent the mean ± SEM of three experiments; * *p* < 0.05; ** *p* < 0.01; **** *p* < 0.0001. (**E**) Scatter plot comparing mRNA levels of TEK, OSMR, PLGF, and CSF1R with those of OLFML3 in tumor samples from CRC patients. Differences were evaluated using Pearson’s correlation and Spearman’s rank correlation analysis. (**F**) Tumor growth curves (upper panel) and ex vivo tumor volume (lower panel) of end-stage MC38 tumors isolated from mice treated with rec9F8 mAb alone or with rec9F8 mAb plus an anti-CSF1-R blocking mAb (*n* = 8–10 per group). Bars represent the mean ± SEM; * *p* < 0.05; *** *p* < 0.001.

**Figure 6 cancers-13-04625-f006:**
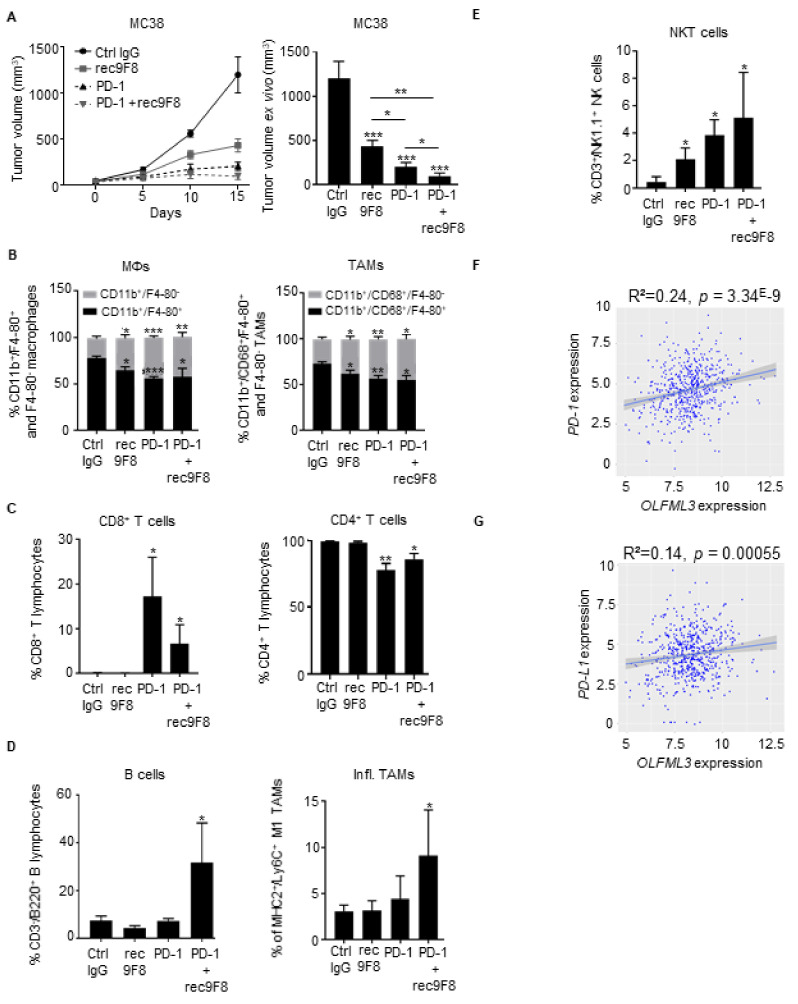
Anti-OLFML3 mAb treatment increases the antitumor effects of anti-PD-1 immunotherapy. (**A**) Growth curve (left) and ex vivo size of MC38 tumors isolated from mice treated with rec9F8 or PD-1 alone or a combination of rec9F8 and anti-PD-1 mAbs at the end of the experiment (right) (*n* = 6–8 per group). Bars (right) represent the mean ± SEM; * *p* < 0.05; ** *p* < 0.01; *** *p* < 0.001. (**B**) Flow cytometry quantification of immature (CD11b^+^/F4-80^−^) and mature (CD11b^+^/F4-80^+^) macrophages (left) and immature (CD11b^+^/CD68^+^/F4-80^−^) and mature (CD11b^+^/CD68^+^/F4-80^+^) TAMs (right) in MC38 tumors isolated from mice treated with control IgG, anti-PD-1, rec9F8, or anti-PD-1 mAbs plus rec9F8. Bars (right) represent the mean ± SEM; * *p* < 0.05; ** *p* < 0.01; *** *p* < 0.001. (**C**) Flow cytometry quantification of intratumoral infiltration of (left) CD8^+^ T lymphocytes (CD45^+^/CD3^+^/CD8^+^) and (right) CD4^+^ T lymphocytes (CD45^+^/CD3^+^/CD4^+^). Bars represent the mean ± SEM; * *p* < 0.05; ** *p* < 0.01. (**D**) Flow cytometry quantification of intratumoral infiltration by (left) B lymphocytes (CD45^+^/CD3^−^/B220^+^) and (right) inflammatory macrophages isolated from mice bearing MC38 tumors treated with rec9F8 and anti-PD-1 mAbs, either alone or in combination as indicated. Bars represent the mean ± SEM; * *p* < 0.05. (**E**) Flow cytometry quantification of intratumoral infiltration by NKT cells (CD45^+^/CD3^+^/NK1.1^+^). Bars represent the mean ± SEM; * *p* < 0.05. (**F**,**G**) Scatter plots with linear regression lines comparing the mRNA levels of PD-1 (**F**) and PD-L1 (**G**) with that of OLFML3 in tumor samples from CRC patients. Differences were evaluated using Pearson’s correlation and Spearman’s rank correlation analyses.

**Figure 7 cancers-13-04625-f007:**
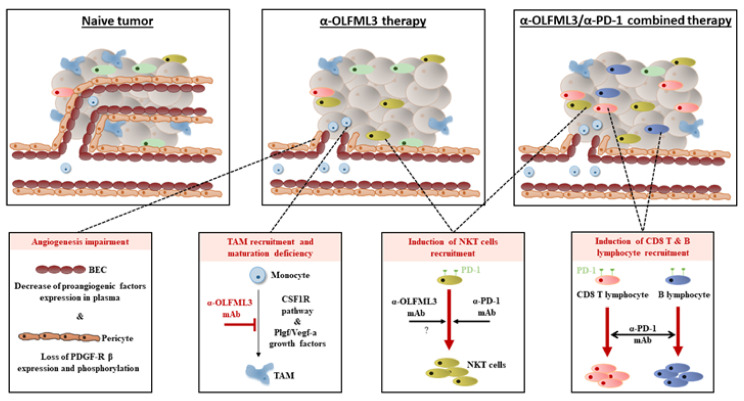
Proposed model illustrating the antitumor effect of anti-OLFML3 antibody treatment alone and in combination with anti-PD-1 checkpoint inhibitors. In naïve tumors, a high number of intratumoral blood vessels and pericytes are observed. Tumors are rich in tumor-associated macrophages but are poorly infiltrated by T, B, and NKT lymphocytes. Targeting of OLFML3 reduces the number of blood vessels, pericytes, and TAMs and increases the number of NKT cells. When combined with an anti-PD-1 checkpoint inhibitor, anti-OLFML3 treatment increases tumor infiltration by B lymphocytes and increases infiltration by NKT lymphocytes. Together, these effects inhibit tumor growth.

## Data Availability

Data are available upon request from the corresponding author.

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
