# Peer review of "Targeting OLFML3 in Colorectal Cancer Suppresses Tumor Growth and Angiogenesis, and Increases the Efficacy of Anti-PD1 Based Immunotherapy"

_cancers, 2021, doi:10.3390/cancers13184625_

Round 1
Reviewer 1 Report
This generally well written manuscript on OLFML3 as a target to anti-angiogenic tumortherapy expecially in combination with checkpoint inhibition nicely combines in vitro, mouse models (both syngenic and xenografted) with data from patient samples. Yet it suffers from several repairable inconsistencies.
Methods – especially animal procedures – in the Methods section are not congruent with the data presented (mainly inconsistent timepoints, but also fixation procedure(s) and mouse strains. This seems to be easily corrected by thorough proof reading.
Some Figures suffer from low resolution. Especially FACS data and the final model (Fig. 7) cannot be displayed properly in the file provided to this reviewer.
Here are this reviewer more detailed comments:
- For comparison, it would be nice to have colon samples included in Supplemental Figure 1 (not only fluorescent stainings in Figure 1)
- Please comment/clarify the contradiction in the use of immunodeficient mouse strains in Materials and Methods: line 124: PDX into CB17 Scid mice vs line 161 (Animal procedures): NSG and l254 (results) NSG
- The tumors of the PDX experiments seem to be quite small (0,15-0,3cm3).
- Do the expression levels shown in Fig. 2A-C relate to human isolates or to PDX (and mouse colon as healthy control)? Please clarify the description.
- Results section, Figure 1 and Suppl. Figure 1 vascular expression density is given as scores. Does that relate only to staining intensity (expression level per cell) or does that score include quantity of postive cells (area?). An explanation of the strong staining in normal kidney and the staining pattern in uterus cancer would be helpfull.
- Tumorgrowth is shown as growth curves and additionally as a bar graph (at experiments end). It seem redundent, since both times it is tumor volume. It would be more informative to give tumor weight at the endpoint (since this corrects for tumor nodes not optimally represented by the standard volume calculation used).
- While the MC38 tumorvolumes are reasonably sized at the end of the experiement, the LLC1 tumors seem very small (especially compared to the data presented in Reference 22, the original paper by the authors).
- In addition a time for tumor growth of only about 1-2 weeks (and even shorter treatment time) maybe a good proof of principle, but not quite usefull for clinical application with tumors quite large at time of diagnosis and the issue tumors overcoming anti-angiogenic treatments (nicely described by the authors in the introduction).
- Times of tumor volume measurements shown in Figures are not consistent with the description in the Methods section.
- In Fig 3C it seems that not only the number of Pecam1 positive cells is different, but also the morphology. This seems to have been recognized by the authors, since the axis of the bar graphs is labelled with ‚structures‘ Please comment on this and correct the typing errors in those axis labellings.
- Control tumors in Figures 3B and E seem to grow differently. Should that be contributed to interexperimental variation or could that be an isotype effect?
- The FACS images in Fig 3D are too small/too low in resolution, that the necessary information cannot be visualized.
- Figure 4A shows PECAM and PECAM/aSMA double staining of LLC1 tumors. The upper row seems redundant information. It woud give more information of pericyte coverage to show single chanels of both stainings (and the overlay) of the same, typical tumor area. Percent positiv area just shows, that both endothelial and aSMA positive cells are reduced, but not in relation to tumor vasculature (expecially since pericytes are not the only aSMA expressing cells). Are there analogue data on the MC38 tumors and on tumors in the knock out mice?
- Figure 4C would profit from putting Vegf-a data separate, therefore permitting better scaling of the other markers.
- Please proof read to correct the few typing and formatting errors.
- Numbering in references section is doubled.
Reviewer 2 Report
In the manuscript entitled “Targeting OLFML3 in colorectal cancer suppresses tumor growth and angiogenesis, and increases the efficacy of immunotherapy” the authors demonstrated that the expression of OLFML3 in colorectal cancer promotes angiogenesis and tumor growth. Moreover, they showed that targeting OLFML3 with a specific antibody increases the anti-tumor effect of anti-PD-1 immunotherapy.
The manuscript is well written but I think that some minor revisions are required:
- The quality of the figures is very poor. Please provide more clear images and rewrite the text.
- The authors analyze the mRNA expression of OLFML3 in CRC PDX. However it would be interested to analyze the expression of this gene at protein levels to confirm its overexpression in metastatic CRC samples compared with healthy tissue.
- Since the expression of OLFML3 in CRCs correlated positively with TGF-ß, it would be interesting to analyze its correlation with L1CAM and CXCR4 which, as demonstrate by Delle Cave and colleagues (doi: 10.7150/thno.54027) are both induced by the tgf-b signaling pathway.
- The authors should confirm that, at protein levels, the bevacizumab treatment significantly reduced expression of OLFML3 and of angiogenesis- and lymphangiogenesis-associated genes MCAM, NG2, and VEGF-C.
